# Factors associated with death in patients with tuberculosis in Brazil: Competing risks analysis

**Paulo Victor de Sousa Viana**[1]ᵒ*, **Natalia Santana Paiva**[2]ᵒ, **Daniel Antunes Maciel Villela**[3]‡, **Leonardo Soares Bastos**[3]‡, **Ana Luiza de Souza Bierrenbach**[4]ᵒ, **Paulo Cesar Basta**[5]ᵒ

**1** Fundação Oswaldo Cruz, Escola Nacional de Saúde Pública Sergio Arouca, Centro de Referência Professor Helio Fraga, Rio de Janeiro, RJ, Brazil, **2** Universidade Federal do Rio de Janeiro, Rio de Janeiro, RJ, Brazil, **3** Fundação Oswaldo Cruz, Programa de Computação Científica, Rio de Janeiro, RJ, Brazil, **4** Hospital Sírio-Libanês; Instituto de Ensino e Pesquisa, São Paulo, SP, Brazil, **5** Fundação Oswaldo Cruz, Escola Nacional de Saúde Pública Sergio Arouca, Rio de Janeiro, RJ, Brazil

ᵒ These authors contributed equally to this work.
‡ These authors also contributed equally to this work.
* paulovictorsviana@gmail.com

**Data Availability Statement:** The data supporting the results of this study are available from the Brazilian National Tuberculosis Program, but restrictions apply to the availability of such data

## Abstract

### Objectives

This study aimed to analyze the factors associated with likely TB deaths, likely TB-related deaths and deaths from other causes. Understanding the factors associated with mortality could help the strategy to End TB, especially the goal of reducing TB deaths by 95% between 2015 and 2035.

### Methods

A retrospective, population-based cohort study of the causes of death was performed using a competing risk model in patients receiving treatment for TB. Patients had started TB treatment in Brazil 2008–2013 with any death certificates dated in the same period. We used three categories of deaths, according to ICD-10 codes: i) probable TB deaths; ii) TB-related deaths; iii) deaths from other causes.

### Results

In this cohort, 39,997 individuals (14.1%) died, out of a total of 283,508 individuals. Of these, 8,936 were probable TB deaths (22.4%) and 3,365 TB-related deaths (8.4%), illustrating high mortality rates. 27,696 deaths (69.2%) were from other causes. From our analysis, factors strongly associated with probable TB deaths were male gender (sHR = 1.33, 95% CI: 1.26–1.40), age over 60 years (sHR = 9.29, 95% CI: 8.15–10.60), illiterate schooling (sHR = 2.33, 95% CI: 2.09–2.59), black (sHR = 1.33, 95% CI: 1.26–1.40) and brown (sHR = 13, 95% CI: 1.07–1.19) color/race, from the Southern region (sHR = 1.19, 95% CI: 1.10–1.28), clinical mixed forms (sHR = 1.91, 95% CI: 1.73–2.11) and alcoholism (sHR = 1.90, 95% CI: 1.81–2.00). Also, HIV positive serology was strongly associated with probable TB deaths (sHR = 62.78; 95% CI: 55.01–71.63).

through the Access to Information Law and are therefore not publicly available. However, the data are made available by the authors upon justifiable request and with authorization from the Brazilian National TB Program. The request should be performed through the URL: http://www.acessoainformacao.gov.br/sistema/site/primeiro_acesso.html.

**Funding:** The authors received no specific funding for this work.

**Competing interests:** The authors declare that they have no competing interests.

## Conclusions

In conclusion, specific strategies for active surveillance and early case detection can reduce mortality among patients with tuberculosis, leading to more timely detection and treatment.

## Introduction

Although there is highly effective treatment, tuberculosis (TB) remains the ninth leading cause of death in the world and the leading cause of death among infectious diseases. In 2018, 1.2 million deaths from TB among seronegative individuals and 251,000 deaths among HIV-positive people were estimated [1]. Although Brazil has considerably reduced incidence and death rates of TB in the last decade, high death rates still pose a real challenge for TB control in the country [2].

Identifying patients at risk of death during TB treatment should be a priority for health surveillance: it is essential for assessing programmatic needs and has the potential to contribute to the targeting of interventions and improvement of treatment monitoring, thus contributing to the End TB Strategy and to reduce TB mortality by 95% [1]. In this sense, understanding the factors associated with treatment failure may allow the development of strategies to assist in a more effective clinical follow-up of the individuals at risk.

Several studies have suggested that factors associated with survival in cases of TB are related to the presence of specific comorbidities, including HIV/AIDS and diabetes mellitus [3–9]; some clinical characteristics, such as drug resistance, default of treatment, extrapulmonary and mixed forms [3, 6, 10, 11]; specific sociodemographic characteristics of patients (age, male gender, schooling, color or race, etc); and lifestyle, such as alcoholism and smoking [4, 7, 11, 12]. In Brazil, few studies of survival analysis after TB treatment were identified in the literature [7, 13]. Existing literature is generally regional in scope and does not address specific TB mortality, nor do they consider TB-related causes and other causes of death independently. It is also worth noting that the factors associated with TB deaths mentioned above have not been studied in any nationwide cohort study.

Therefore, this study aimed to analyze the factors associated with probable TB deaths, TB-related deaths and deaths from other causes, using competing risk model in a cohort of patients receiving treatment for TB from 2008 to 2013, across Brazil.

## Methods

### Study population

This is a retrospective cohort study including all patients who initiated treatment for TB in Brazil from January 1, 2008, to December 31, 2013.

Two national data sources were used: The Notifiable Diseases Information System (SINAN, acronym in Portuguese) and the Mortality Information System (SIM). The SINAN database contained cases reported between January 1, 2008, and December 31, 2011, and was extracted on 09/20/2013. The SIM database contained deaths reported between January 1, 2008, and December 31, 2013, and was extracted on 04/01/2016. Both databases had nominal information.

The study population consisted of all new cases of tuberculosis reported in SINAN that began treatment in the period from January 1, 2008 (date of first entry) until December 31, 2011 (date of last entry). Through record linkage of data with the SIM database, the cases were

followed up until the occurrence of deaths or until December 31, 2013, when administrative censoring was considered (end of the follow-up period).

In order to guarantee the quality of information on TB treatment episodes, an automatic monitoring method adopted by Bierrenbach et al. [14] sought to eliminate duplicates and correct classification errors of different treatment episodes from the same patient. Thus, we excluded true duplications (records of the same patient by the same health unit and the same date of initiation of treatment, only the oldest, or most complete, if both had the same notification date kept). The cases classified as transference in the variable type of entry and the missing information were corrected. When the first entry was classified as "do not know" correct for a new case. We excluded records of cases terminated as a "change in diagnosis" (i.e., not TB), to analyze only the new cases in the 1st treatment entry. Therefore, cases classified as return after default, relapse, and transfer were excluded. We also excluded inconsistencies in treatment starting dates and date of outcome (i.e., cases with treatment date after date of outcome), as well as missing dates (Fig 1).

This study was approved by the Research Ethics Committee of the National School of Public Health/FIOCRUZ, under the protocol: CAAE: 14643713.0.0000.5240. The nominal identifiers were removed from the database after the data linkage, ensuring the privacy of the subjects involved in the study.

No informed consent was obtained since only the secondary notification data were analyzed.

## Study definitions

SINAN, linked to SIM has been used to evaluate tuberculosis outcomes [15, 16]. SINAN and SIM are fed continuously, have good coverage and quality, although the coverage is heterogeneous within the country, with large variations among the states. Moreover, delays in data processing, under-reporting of cases and deaths, variation of the quality and coverage in different geographical areas, as well as incorrect filling of death certificates, are some of the limitations [17].

In Brazil, the National Tuberculosis Control Program (NTCP) uses the following definition to case of tuberculosis: any individual with a diagnosis confirmed by smear microscopy or culture, and one in which the physician, based on clinical-epidemiological data and the result of complementary tests, makes the diagnosis of tuberculosis [18].

The outcome categories of the study were created for the causes of death according to ICD-10 codes: i) probable TB deaths, those that had underlying cause with codes A15 to A19 of ICD-10, 'probable' is used as definition relies on death certificate and not autopsy; ii) TB-related deaths, those in which there was mention of any of the ICD-10 codes (A15-A19), referring to TB in any line of part 1 of the death certificate; iii) death with no mention of TB, those deaths in which there was no mention of TB (codes A15-A19 of ICD-10) in any part of the death certificate, considered here as other causes of deaths.

Based on the literature review, we tested the following covariables as associated factors with probable TB deaths: sex (female/male); schooling (illiterate, under 8 years old, over 8 years old and ignored); age group (0 to 19 years, 20 to 39 years, 40 to 59 years and 60 or more); color or race (white, black, brown, yellow, indigenous and ignored); macro-region of residence (North, Northeast, Southeast, South, and Central-West); clinical forms (pulmonary, extrapulmonary and mixed); number of treatments (1; 2 to 3; 4 or more); HIV serology (positive, negative, in progress and not performed); alcoholism (yes and no); diabetes (yes and no). The variable number of treatments refers to the number of episodes that a patient underwent treatment during the follow-up period.

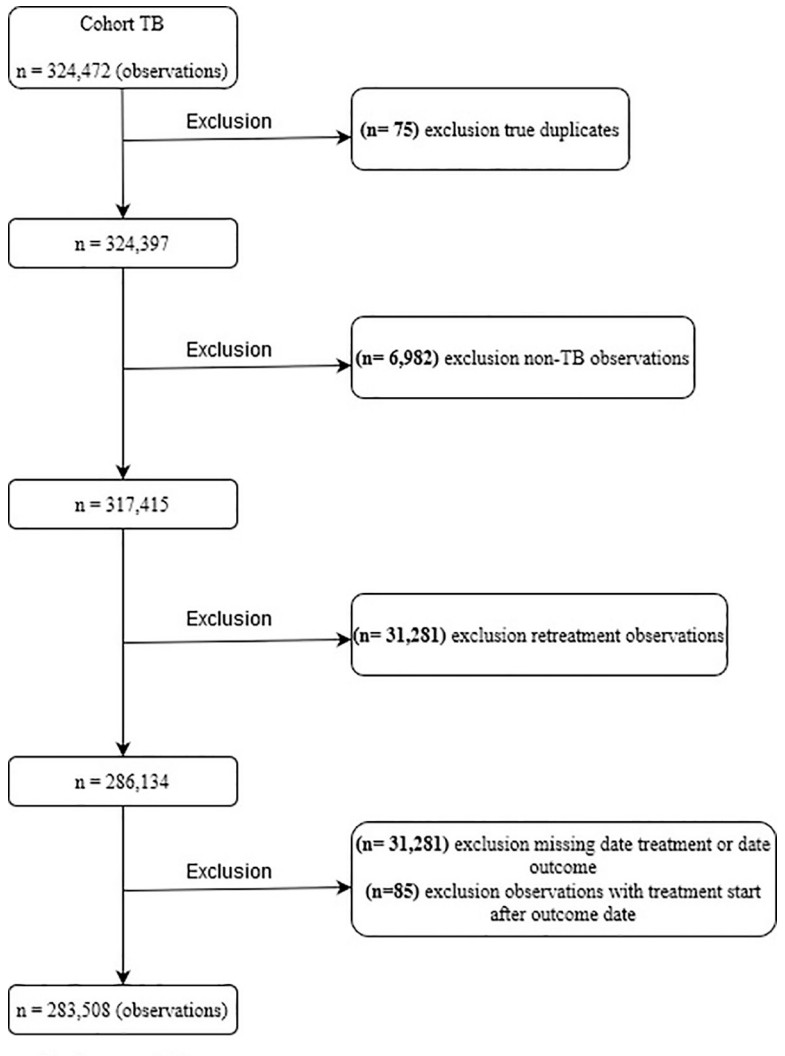

**Fig 1. Flowchart illustrating criterion selection cases in Brazil, 2008 to 2013.**

The variable color or race, according to the categories adopted by the Brazilian Geography and Statistics Institute (IBGE) was introduced in information systems managed by the Ministry of Health from 2000. In practice, the color/race variable in SINAN notification form is reported from the patient's self-declaration, based on the color of their skin, according to the five terms used by the IBGE: white, black, brown, yellow, and indigenous [19, 20].

The main tool of the SINAN database is the notification/investigation form that includes both individual data of the patients as well as epidemiological and follow-up data until the end of treatment, on average 6 to 9 months. This information must be sent regularly from the local health units, since the municipality level, to the Ministry of Health, in order to produce the TB monitoring bulletin [21, 22].

SIM is the oldest health information system in the country, created in 1975 by the Ministry of Health to address civil registry failures. It is a system with high population coverage, which aims to record data on mortality in Brazil, comprehensively, and reliably. Currently, SIM's coverage is estimated at more than 95% in Brazil [23]. Data available in the SIM is essential to

understand the mortality profile of a population. It is used to calculate health indicators, perform trend analysis and to establish investment priorities in the health sector.

Death certificates are the fundamental sources for SIM. Adequate completion of the death certificate, which must necessarily be performed by physicians [24], is an essential condition for good information quality of the SIM data. Information registered in the death certificate allows adequate knowledge of the causes of death of an individual, if well completed. The term underlying cause, as defined by WHO in successive revisions of the International Statistical Classification of Diseases and Related Health Problems (ICD), refers to the "cause of death" that initiated the sequence of morbid events that led the individual to die. In addition to the underlying cause, the associated causes, which include the terminal and intermediate causes resulting from the underlying cause, as well as the causes that contributed to death without direct relation to the pathological process responsible for it, are also recorded in the death certificate.

## Linkage procedures

According to the Brazilian legislation regulating access to secondary data [25], we obtained authorization from the Ministry of Health for the use of nominal identification data and, therefore, could perform record linkage procedures between SINAN and SIM databases for the study period.

The database linkage between SINAN and SIM was performed in a three-step process. The first one was conducted in SINAN's database using a deterministic algorithm for semi-automatic linking records, similar to those validated by Pacheco et al. [26] and Oliveira et al. [27], with an adaptation to the STATA statistical software. The second step was the linkage between the databases of SINAN and SIM. We used a probabilistic data linkage using a methodology called Bloom filter [28]. For this linkage, the following vital fields were employed: patient's name, mother's name, date of birth, and code of the municipality of residence for the purpose of identifying pairs of the same individual using the free software R 3.1.2 and package "PPRL" [29]. Finally, to increase the specificity of the matches found using the second step, a deterministic linkage was performed with an algorithm similar to the step one. To obtain more details on the database linkage process, see S1 File.

## Statistical analysis

The four study groups [probable TB deaths, TB-related deaths, death with no mention of TB, and no death (censoring) reported until December 31, 2013] were compared in a descriptive analysis regarding the variables of interest.

Survival analysis was used to elucidate factors associated with probable TB deaths (TB as the underlying cause) considering the presence of competing risks. For this analysis, the "censoring" outcome category was used as a reference for the outcome response variable, being compared with censoring vs. probable TB deaths; censoring vs. TB-related death and censoring vs. death with no mention TB. The Fine & Gray sub-distribution model based on the cumulative incidence function (CIF) was used as a reference [30], considering the probability of an event occurring before a specific time. This model considers a proportional risk model for the sub-distribution of competing risk, where the covariates directly affect the CIF. Thus, the observations on competing events should be maintained in the range of risks. That is, for individuals in our study that died due to other causes rather than TB as the underlying cause, the model considers these risks but with a decreasing weight to take into account the reduction of the observations [31].

Survival time was measured in days between the date of the beginning of first treatment and the date of the end of treatment or of the event of interest [probable TB death, TB-related

death, death with no mention of TB or censoring (end of follow-up on 12/31/2013)]. On the other hand, from the deterministic linkage of the data, the fatal outcomes were divided into three groups (above mentioned) of analysis for competing events according to the ICD-10 codes listed in part I of the death certificates and made available in the SIM.

The cumulative incidence function was used to describe the probability of TB mortality in the presence of competing events and the Gray test was used to compare the differences between the groups. The Fine-Gray subdistribution hazard model was used to identify factors associated with mortality among TB cases. Based on a list of selected variables that showed previous association with death due to TB (presented above), we tested each of them individually, verifying its statistical significance, considering a p-value <0.20. The variables that had a p-value lower than 0.20 in the univariate analysis were selected as a candidate for the Fine & Gray multiple model. In the final model, we considered the level of significance of 5% (p<0.05). The risk measure was the subdistribution hazard ratio (sHR) with its respective 95% confidence intervals. The proportionality assumption of the Fine-Gray model was initially checked for CIF and Schoenfeld residuals tests.

### Softwares used

Microsoft Excel spreadsheets 2016 were used to structure the data (Microsoft Corp., Redmond, WA, USA). We conducted the statistical analysis in STATA software, College Station, TX, USA [32], and free software R version 3.3.2 (R Foundation for Statistical Computing, Vienna, Austria) in the "Survival" [33] and "Riskregression" packages [34].

## Results

### Study population

During the study period, a total of 283,508 subjects met inclusion criteria, including 39,997 (14.1%) who died. A number of 8,936 deaths (22.4%) were attributed to tuberculosis as the probable cause, 3,365 deaths (8.4%) had tuberculosis as an associated cause, and 27,696 deaths (69.2%) had no mention of tuberculosis in the death certificate. Together, probable TB deaths and TB-related deaths made up 30.8% of the total deaths, demonstrating a high mortality rate in patients receiving TB treatment in Brazil, between 2008–2013. The median follow-up of the entire cohort was 1,348 days (IQR: 949–1,761). The median elapsed time since treatment start for probable TB deaths was 27 days (IQR: 5–126), for TB-related deaths was 52 days (IQR: 9–196) and among deaths for other causes/with no mention of TB was 383.5 days (IQR: 73–875).

### Sociodemographic and clinical characteristics

The majority of deaths (29,322, 15.7%) in our analysis occurred in males, regardless of the cause. The majority of the cases occurred among patients aged 20 to 39 years old. Higher death proportions were observed among the oldest age group for probable TB deaths and deaths with no mention of TB (8.9% and 23.0%, respectively). There were higher proportions of probable TB deaths (5.2%) and deaths with no mention of TB (12.9%) among the patients who had no schooling (Table 1).

Death proportions were lower among self-reported indigenous cases (2.8% for probable TB deaths, 0.3% for TB-related deaths, and 5.3% for deaths with no mention of TB). Most cases were from the Southeast region 129,874 (45.8%); however, higher proportions of probable TB deaths were from the Central-West region (3.6%). The Southern region had higher death rates with TB as an associated cause and deaths with no mention of TB (2.2% and 11.8%, respectively) (Table 1).

**Table 1. Sociodemographic and clinical characteristics of study participants, Brazil, 2008 to 2013 (n = 283,508).**

| Variables | Censoring | | Likely TB-related deaths | | Death associated with TB | | Death with no mention of TB (other causes) | | Total | |
|---|---|---|---|---|---|---|---|---|---|---|
| Follow-up time in days[a] | 1,450 (1,079–1,817) | | 27 (5–126) | | 52 (9–196) | | 383,5 (73–876 | | 1,348 (949–1,761) | |
| **Sex** | n | % | n | % | n | % | n | % | n | % |
| Male | 157,455 | 84.3 | 6,729 | 3.6 | 2,306 | 1.2 | 20,287 | 10.9 | 186,777 | 65.9 |
| Female | 86,056 | 89.0 | 2,207 | 2.3 | 1,059 | 1.1 | 7,409 | 7.7 | 96,731 | 34.1 |
| **Age group (years)** | | | | | | | | | | |
| 0 a 19 | 25,930 | 96.2 | 243 | 0.9 | 71 | 0.3 | 699 | 2.6 | 26,943 | 9.5 |
| 20 a 39 | 115,640 | 91.6 | 1,693 | 1.3 | 1,631 | 1.3 | 7,341 | 5.8 | 126,305 | 44.5 |
| 40 a 59 | 76,544 | 82.8 | 3,620 | 3.9 | 1,342 | 1.5 | 10,941 | 11.8 | 92,447 | 32.6 |
| 60+ | 25,397 | 67.2 | 3,380 | 8.9 | 321 | 0.8 | 8,715 | 23.0 | 37,813 | 13.4 |
| **Schooling** | | | | | | | | | | |
| Illiterate | 16,112 | 81.0 | 1,029 | 5.2 | 174 | 0.9 | 2,566 | 12.9 | 19,881 | 7.0 |
| Less 8 years | 91,104 | 85.6 | 3,192 | 3.0 | 1,346 | 1.3 | 10,787 | 10.1 | 106,429 | 37.5 |
| Greater 8 years | 46,369 | 92.5 | 560 | 1.1 | 343 | 0.7 | 2,837 | 5.7 | 50,109 | 17.7 |
| Ignored | 89,926 | 84.0 | 4,155 | 3.9 | 1,502 | 1.4 | 11,506 | 10.7 | 107,089 | 37.8 |
| **Color or race** | | | | | | | | | | |
| White | 83,321 | 85.3 | 2,885 | 3,0 | 1,157 | 1.2 | 10,268 | 10.5 | 97,631 | 34.4 |
| Black | 30,569 | 85.2 | 1,187 | 3,3 | 541 | 1.5 | 3,580 | 10.0 | 35,877 | 12.6 |
| Brown | 98,499 | 86.8 | 3,578 | 3,2 | 1,227 | 1.1 | 10,168 | 9.0 | 113,472 | 40.0 |
| Yellow | 2,336 | 87.2 | 85 | 3,2 | 21 | 0.8 | 238 | 8.9 | 2,680 | 0.9 |
| Indigenous | 2,941 | 91.6 | 89 | 2,8 | 11 | 0.3 | 171 | 5.3 | 3,212 | 1.1 |
| Ignored | 25,845 | 84.4 | 1,112 | 3,6 | 408 | 1.3 | 3,271 | 10.7 | 30,636 | 10.8 |
| **Region** | | | | | | | | | | |
| North | 25,123 | 88.2 | 727 | 2.6 | 232 | 0.8 | 2,409 | 8.5 | 28,491 | 10.1 |
| Northeast | 67,368 | 86.9 | 2,560 | 3.3 | 611 | 0.8 | 6,964 | 9.0 | 77,503 | 27.3 |
| Southeast | 111,411 | 85.8 | 4,153 | 3.2 | 1,563 | 1.2 | 12,747 | 9.8 | 129,874 | 45.8 |
| South | 29,231 | 83.0 | 1,048 | 3.0 | 775 | 2.2 | 4,143 | 11.8 | 35,197 | 12.4 |
| Central-West | 10,378 | 83.4 | 448 | 3.6 | 184 | 1.5 | 1,433 | 11.5 | 12,443 | 4.4 |
| **Number of treatments** | | | | | | | | | | |
| 1 | 234,851 | 86.1 | 8,514 | 3.1 | 3,065 | 1.1 | 26,473 | 9.7 | 272,903 | 96.3 |
| 2 a 3 | 8,473 | 81.8 | 404 | 3.9 | 283 | 2.7 | 1,198 | 11.6 | 10,358 | 3.7 |
| 4 or more | 187 | 75.7 | 18 | 7.3 | 17 | 6.9 | 25 | 10.1 | 247 | 0.1 |
| **Clinical form** | | | | | | | | | | |
| Pulmonary | 202,510 | 86.6 | 7,640 | 3.3 | 2,227 | 1.0 | 21,603 | 9.2 | 233,980 | 82.5 |
| Extrapulmonary | 34,191 | 84.8 | 876 | 2.2 | 654 | 1.6 | 4,581 | 11.4 | 40,302 | 14.2 |
| Mixed forms | 6,810 | 73.8 | 420 | 4.6 | 484 | 5.2 | 1,512 | 16.4 | 9,226 | 3.3 |
| **HIV serology** | | | | | | | | | | |
| Positive | 17,442 | 64.9 | 319 | 1.2 | 2,703 | 10.1 | 6391 | 23.8 | 26,855 | 9.5 |
| Negative | 127,027 | 90.1 | 3,453 | 2.4 | 251 | 0.2 | 10211 | 7.2 | 140,942 | 49.7 |
| In progress | 20,663 | 88.6 | 655 | 2.8 | 92 | 0.4 | 1901 | 8.2 | 23,311 | 8.2 |
| Not performed | 78,379 | 84.8 | 4,509 | 4.9 | 319 | 0.3 | 9193 | 9.9 | 92,400 | 32.6 |
| **Alcoholism** | | | | | | | | | | |
| Yes | 30,302 | 79.0 | 2,241 | 5.8 | 669 | 1.7 | 5,126 | 13.4 | 38,338 | 13.5 |
| No | 213,209 | 87.0 | 6,695 | 2.7 | 2,696 | 1.1 | 22,570 | 9.2 | 245,170 | 86.5 |
| **Diabetes** | | | | | | | | | | |
| Yes | 13,174 | 78.2 | 826 | 4.9 | 122 | 0.7 | 2,718 | 16.1 | 16,840 | 5.9 |

*(Continued)*

**Table 1.** (Continued)

| Variables | Censoring | | Likely TB-related deaths | | Death associated with TB | | Death with no mention of TB (other causes) | | Total | |
|---|---|---|---|---|---|---|---|---|---|---|
| No | 230,337 | 86.4 | 8,110 | 3.0 | 3,243 | 1.2 | 24,978 | 9.4 | 266,668 | 94.1 |

<sup>a</sup> median (IQR)

Almost all cases (96.3%) had been subject to a single treatment. However, higher proportions of probable TB deaths and TB-related deaths were observed among individuals with multiple treatments (4 or more) when compared to those with a single or with 2–3 treatments (7.3% and 6.9%, respectively). While the pulmonary form was the most frequent one (82.5%) of the total cohort, cases with "mixed forms" proved most fatal (4.6% for probable TB deaths, 5.2% for TB-related deaths, and 16.4% for deaths with no mention of TB). HIV serology was more frequent among "TB-related deaths" and "deaths with no mention of TB" (10.1% and 23.8%), being less frequent among probable TB deaths (4.9%) (Table 1).

For 13.5% of patients, alcohol abuse was identified, with the highest proportion of alcoholism among those that died with no mention of TB (13.4%) and among deaths that had TB as the underlying cause (5.2%). The presence of diabetes mellitus was reported in 5.9% of patients and was more frequent among deaths with no mention of TB (16.1%) and probable TB deaths (4.9%) (Table 1).

## Survival analysis

The cumulative incidence functions for the risk sub-distribution ratio (sHR) of causes of probable TB death, TB-related death and death with no mention of TB in a competing risk structure are presented in Fig 2. The cumulative incidence of causes of death with no mention of TB was higher than non-TB related causes and associated TB throughout the observation period, with a gradual increase of TB during the follow-up. We also estimated the probabilities of cumulative incidence of deaths for patients stratified according to HIV serology results. A significant steady and gradual increase was observed among deaths without mention of TB (other causes) (see Fig 3 in S2 File).

Table 2 shows the final model of the risk sub-distribution ratio (sHR) and the 95% confidence interval (95% CI) for the covariates selected for the different causes of mortality among

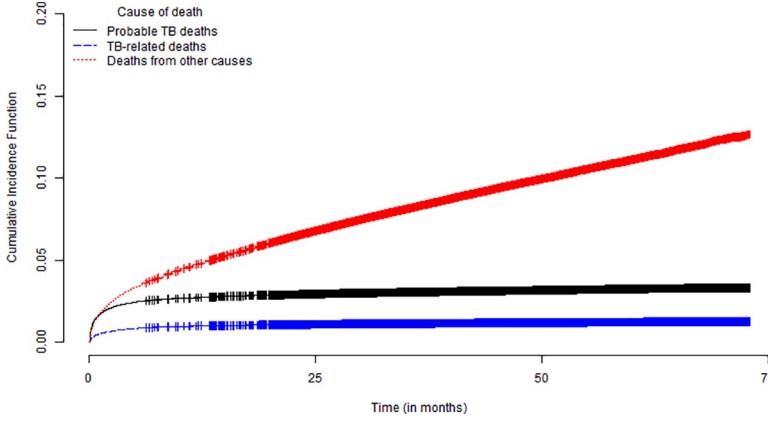

**Fig 2. Survival curves estimated by the Cumulative Incidence Functions (CIF) of the sub-distributions of risks proposed by Fine-Gray of the deaths.**

**Table 2. Fine-Gray final model with the ratios of risk sub-distributions (sHR) and confidence interval (CI 95%) of reported cases of tuberculosis according to different causes of deaths (censoring vs probable TB death; censoring vs TB-related death; and censoring vs death with no mention TB), Brazil, 2008 to 2013 (n = 283,508).**

| Cause of death | Likely TB-related deaths | | Death associated with TB | | Death with no mention of TB (other causes) | |
|---|---|---|---|---|---|---|
| Variables | sHR[a] | IC 95% | sHR | IC 95% | sHR | IC 95% |
| **Sex** | | | | | | |
| Male | 1.33 | 1.26–1.40*** | 0.94 | 0.87–1.01 | 1.29 | 1.25–1.32*** |
| Female | Ref [e] | | Ref | | Ref | |
| **Age group (years)** | | | | | | |
| 0 a 19 | Ref | | Ref | | Ref | |
| 20 a 39 | 1.53 | 1.34–1.76*** | 2.09 | 1.64–2.66*** | 1.89 | 1.74–2.03*** |
| 40 a 59 | 3.95 | 3.46–4.50*** | 2.53 | 1.98–3.22*** | 3.87 | 3.58–4.18*** |
| 60+ | 9.29 | 8.15–10.60*** | 4.26 | 3.29–5.52*** | 10.36 | 9.59–11.20*** |
| **Schooling** | | | | | | |
| Illiterate | 2.33 | 2.09–2.59*** | 1.72 | 1.43–2.08*** | 1.64 | 1.55–1.73*** |
| Less 8 years | 1.74 | 1.59–1.91*** | 1.56 | 1.38–1.76*** | 1.41 | 1.35–1.47*** |
| Greater 8 years | Ref | | Ref | | Ref | |
| Ignored | 2.57 | 2.35–2.81*** | 1.73 | 1.53–1.96*** | 1.58 | 1.51–1.65*** |
| **Color or race** | | | | | | |
| White | Ref | | Ref | | Ref | |
| Black | 1.10 | 1.02–1.18** | 1.29 | 1.16–1.43*** | 0.98 | 0.94–1.01 |
| Brown | 1.13 | 1.07–1.19*** | 1.24 | 1.13–1.35*** | 0.96 | 0.93–0.99*** |
| Yellow | 1.01 | 0.82–1.26 | 1.22 | 0.79–1.89 | 0.92 | 0.80–1.04 |
| Indigenous | 0.96 | 0.77–1.19 | 0.70 | 0.39–1.28 | 0.59 | 0,51–0,69*** |
| Ignored | 1.07 | 0.99–1.16 | 1.18 | 1.05–1.33** | 0.99 | 0.95–1.03 |
| **Region** | | | | | | |
| North | 0.81 | 0.75–0.89*** | 0.85 | 0.73–0.98* | 0,95 | 0.91–1.00 |
| Northeast | 0.84 | 0.80–0.89*** | 0.85 | 0.77–0.94** | 0,89 | 0.86–0.92*** |
| Southeast | Ref | | Ref | | Ref | |
| South | 1.19 | 1.10–1.28*** | 1.21 | 1.10–1.34*** | 1,08 | 1.04–1.12*** |
| Central-West | 1.07 | 0.97–1.64 | 1.40 | 1.20–1.64*** | 1,16 | 1.10–1.23*** |
| **Clinical form** | | | | | | |
| Pulmonary | Ref | | Ref | | Ref | |
| Extrapulmonary | 0.82 | 0.76–0.88*** | 0.92 | 0.85–1.01 | 1.15 | 1.11–1.19*** |
| Mixed forms | 1.91 | 1.73–2.11*** | 1.62 | 1.46–1.79*** | 1.41 | 1.34–1.49*** |
| **HIV serology** | | | | | | |
| Positive | 0.59 | 0.53–0.67*** | 62.78 | 55.01–71.63*** | 4.44 | 4.29–4.58*** |
| Negative | Ref | | Ref | | Ref | |
| In progress | 1.17 | 1.08–1.28*** | 2.27 | 1.78–2.88*** | 1.11 | 1.05–1.16*** |
| Not performed | 2.00 | 1.91–2.10*** | 2.01 | 1.70–2.38*** | 1.31 | 1.27–1.35*** |
| **Alcoholism** | | | | | | |
| Yes | 1.90 | 1.81–2.00*** | 1.38 | 1.27–1.51*** | 1.35 | 1.31–1.38*** |
| No | Ref | | Ref | | Ref | |

Ref: Reference category

***$p$-valor<0.001

**$p$-valor<0.01

*$p$-valor<0.05

[a] as mentioned in the methods.

the reported cases of tuberculosis. The covariates that correlated with death were sex, age, schooling, color or race, macro-region, clinical form, HIV serology, and alcoholism.

The Fine-Gray models indicated that male subjects had a higher risk of probable TB death TB (sHR: 1.33 95% CI: 1.26–1.40) and deaths with no mention of TB (sHR: 1.29 (95% CI: 1.25–1.32). Patients over 60 years of age were more likely to die, irrespective of cause. For probable TB deaths and for those with no mention of TB, the sHR, as compared to the age group of 0–19 years was 9.29 (95% CI: 8.15–10.60) and 10.36 (95% CI: 9.59–11.20) respectively. The Ignored and illiterate schooling categories were strongly associated with death irrespective of cause, presenting a risk gradient, as schooling declined. Regarding the risk of death among color or race categories, browns had a higher risk of probable TB death (sHR = 1.13 95% CI: 1.07–1.19). Among the TB-related deaths, the highest risk was among blacks (sHR = 1.29, 95% CI: 1.16–1.43), and a lower risk of deaths with no mention of TB was observed among indigenous and brown persons (sHR = 0.59, 95% CI: 0.51–0.69 and sHR = 0.96, 95% CI: 0.93–0.99, respectively) (Table 2).

Patients from the South and Central-West regions of the country had higher risks of probable TB death (sHR = 1.19 95% CI: 1.19–1.28), while TB-related death and deaths with no mention of TB (sHR = 0.59, 95% CI: 0.51–0.69 and sHR = 0.96, 95% CI: 0.93–0.99, respectively) were less frequent in that regions. The mixed clinical form presented higher risk of death from all causes (sHR = 1.91, 95% CI: 1.73–2.11, sHR = 1.62, 95% CI: 1.46–1.79; sHR = 1.41; 95% CI: 1.34–1.49, for probable TB death, TB-related death and death with no mention of TB, respectively) (Table 2).

HIV positive serology showed a higher risk of death for TB-related death (sHR = 62.78 CI95%: 55.01–71.63) as compared to HIV negative individuals. On the other hand, positive serology was a protective factor for probable TB death (sHR = 0.59 95% CI: 0.53–0.67). Individuals with alcoholism comorbidity presented a higher risk among probable TB deaths (sHR = 1.90 CI95%: 1.81–2.00) as compared to those without this comorbidity.

## Discussion

This study is the first to draw on Brazil's national databases to investigate factors associated with death in patients receiving treatment for TB, taking into account the presence of competing events in survival analysis. While studies usually look at survival analysis with only two categories, probable TB deaths and deaths from other causes, this study includes TB-related deaths as a third category of analysis. Our findings indicate that the main factors associated with deaths, regardless of the cause, include age group, schooling, mixed clinical form, HIV serology and alcoholism.

There was a dose-response effect in the risk of mortality for differing age groups from 20 years old, with a significant risk for patients over 60, particularly in deaths from other causes and probable TB deaths. Looking at schooling, illiterate patients presented the highest risk of mortality, the effect being more evident in probable TB deaths. The association of mixed clinical form with higher rates of death with any cause likely reflects the clinical severity of the cases, particularly among those where treatment began shortly before the patient died as a result of probable TB. Within HIV serology however, the data are contradictory. While positive HIV serology was strongly associated with TB-related deaths and deaths from other causes, probable TB deaths often presented with HIV serology in progress or not performed. It could be that these inconclusive HIV serology tests act as an indicator of the poor performance of the relevant health service. Finally, alcoholism was associated with all forms of mortality, most strongly associated with probable TB death.

The Fine-Gray model, considering the presence of competing risks, revealed significant changes in hazard risks when considering three different groups of causes of death (probable

TB deaths, TB-related deaths and deaths from other causes) among patients undergoing treatment for TB in a cohort of approximately 300,000 people throughout the national territory.

We understand that classical survival analysis is not appropriate when analyzing time elapsed from an initial event to the occurrence of complex events, or in this case competing events, where the individual is at risk of more than one cause of death. A possibility for analyzing data in the presence of competing risks is the Fine-Gray sub-distribution hazard model of risk that has proved useful in the analysis of the factors associated with death for cases receiving treatment for TB in Brazil.

Death due to TB is considered a preventable sentinel event. The disease has a straightforward diagnosis, there are free medications available in the public health network throughout Brazil, a complete treatment with first-line drugs is relatively inexpensive and the disease is curable in almost 100% of cases. Therefore, the high number of deaths analyzed here points to weaknesses in the Brazilian model of care offered to these patients. These failures range from difficulties in access to diagnosis and treatment in the primary care services to access to emergency services and hospitalization for patients with advanced stages of illness [35].

As reported elsewhere [36–38], in this study we identified a high risk of death during treatment, especially among men. Among the explanations for the additional risk of death in men is the lower predisposition to adherence to treatment regimens [39] and a lifestyle that includes excessive use of alcohol and tobacco, as well as malnutrition when compared to women [40, 41].

In this analysis, individuals aged 60 years or older had a higher chance of death in the different causes analyzed. Similar results have been described by other authors [37, 42] The fact that suffering from TB in older age carries a significantly higher risk of death should be considered when following up these patients throughout their treatment. The coexistence of other diseases related to old age may be associated with late diagnosis of TB due to atypical clinical presentation of the symptoms or the development of extra-pulmonary forms difficult to diagnose. This may well complicate the clinical picture and increase the risk of death.

Interestingly, although the incidence of TB in indigenous populations is consistently higher than that observed among the general population, as reported by several authors [43–45], both the number (absolute and proportional) and the risk of death among self-classified indigenous patients were lower than other color or race categories. Thus, this finding is consistent with other studies that analyzed SINAN's reporting data, with emphasis on the variable outcome, in local or regional contexts [44–46]. On the other hand, the data may be hiding difficulties for an adequate follow-up of TB cases under treatment, especially in the most isolated areas of the country, such as the Amazon region, where most of the Indigenous Lands are found.

Although the concept of race is widely recognized as a social construct, it represents an important indicator of health and is mostly based on a social definition of race, rather than a biological or genetic characterization. It is worth noting the higher chances of probable TB death and TB-related death among individuals classified as "black" and "brown" as compared to "white". The identification of racial groups at higher risk is crucial for National Tuberculosis Control Programs which should lead to the development of differential strategies of follow-up, in an attempt to close the historical, sociological, socioeconomic, and access to health services gaps [20, 47].

TB is the most common opportunistic infection, and several studies have shown that it is the leading cause of death among people living with HIV/AIDS [48, 49], accounting for approximately 25% of HIV/AIDS deaths. In this analysis, similar to other studies [12, 42], a 62-fold higher risk of TB death was identified among seropositive patients who started TB treatment. Thus, the early discovery of seropositivity is a potent weapon against tuberculosis in HIV positive individuals, the early initiation of antiretroviral therapy is an important protective factor against the development of TB-disease in people living with HIV (reduced risk of

tuberculosis among Brazilian patients with advanced human immunodeficiency virus infection treated with highly active antiretroviral therapy). The fact that in our cohort there were 319 HIV positive individuals that died due to TB and not due to AIDS with TB as an associate cause, probably reflects misclassification of cause of death errors, however, this number does not change the risk estimates of the analysis due to its low frequency of events.

People infected with the human immunodeficiency virus (HIV) have 20–30 times greater risk of developing tuberculosis than those without HIV. Thus, the need for treatment of latent TB infection has increased due to its high frequency in people with HIV, if treated prophylactically they can prevent outcomes such as death [50]. A systematic review carried out in 2010, including a total of 12 randomized controlled trials, demonstrated an important reduction in mortality and active incidence of TB in people living with HIV after chemoprophylactic therapy with isoniazid [51].

The prevalence of alcohol abuse disorders in Brazil is around 13.7% in the adult population [52]. In our study, based on information contained in the notification forms, the prevalence of alcoholism among patients reached 13.5%. Alcohol abuse has been recognized as a relevant risk factor for death among cases starting treatment for TB [53, 54].

Record linkage between SINAN and SIM databases is a widely used strategy to identify the underreporting of TB cases and TB deaths in Brazil [14, 55]. In our study, this method allowed us to identify deaths that were not captured in the follow-up of the treatment cohort, as previously reported by other authors [55, 56]. However, as mentioned in the methodology section, we have had access only to death records that had been linked to TB treatment records, after an initial probabilistic linking stage with high sensitivity/low specificity that was performed by the Ministry of Health staff. Therefore, we have had no access to the total amount of deaths that occurred in Brazil between 2008 and 2013 and due to this we were not able to estimate the underreporting of TB cases. This limitation impairs recovering deaths that were probably caused by TB or associated with TB as reported in the death certificate but not reported into the SINAN. In other words, it was only possible to seek complementary information in the death certificate of patients enrolled in the TB notification registry (SINAN) [57].

Despite the illustrative findings and the national scope of this study, it is important to take some limitations into account. As with any study based on reporting secondary data, there may have been underreporting of cases and/or deaths, the incompleteness of some variables, errors in classification, and/or coding of cases. Misclassification errors in particular may have occurred for causes of death in the death certificates and with variables color or race. Another important limitation is that, since it is a study of secondary databases, there is the possibility of residual confounding factors (smoking, COPD, etc.) that are not identified or not measured. Thus, causal inferences cannot be made.

In our analyses, we included 266,668 cases. On the one hand, the survival estimates presented here, in the presence of competing events, identified significant differences in death risks with notable statistical significance and narrow confidence intervals. On the other hand, due to the large size of the sample, the standard errors became extremely small; however, the associations found in this study show biological plausibility and are widely recognized in the literature, so that we do not consider our results to be unfeasible.

Finally, our findings indicate that probable TB deaths or TB-related deaths occurred in the first weeks of treatment, with median times of 27 and 52 days, respectively, while deaths from other causes occurred at a median of more than a year after treatment started when most patients had finished their treatment. This result indicates that patients were probably diagnosed with TB late and at an advanced stage of the disease. This situation is perhaps related to the high rate of mortality reported here. Early diagnosis is key to a favourable outcome as established in the literature [57].

Probable TB deaths and TB-related deaths were strongly associated with males, age over 60, illiterate people, individuals of black/brown color, those who presented with mixed clinical forms and were patients with HIV and suffering from alcoholism. It is therefore possible to assume that if there was active surveillance of cases being treated, together with strategies for early detection and adequate follow-up of the patients, mortality due to tuberculosis could be reduced.

## Conclusions

In conclusion, understanding the factors associated with mortality in patients receiving TB treatment may be a useful contribution to the End TB strategy, particularly in working towards the target to reduce TB deaths by 95%, between 2015 and 2035. It could also help to organize local health services to offer optimal support to patients and to prevent avoidable deaths.

## Supporting information

**S1 File.**
(DOCX)

**S2 File.**
(RAR)

## Acknowledgments

The Brazilian Ministry of Health and the National Tuberculosis Control Program (PNCT) for the availability of data, especially Maria de Fatima Marino de Souza, Denise Arakaki-Sanchez, Patricia Barthlomay and Danielle Pelissari. Noteworthy is the important support of Antony Stevens in one of the stages of database linkage. We also thank MD Joseph Kempton of the University Hospitals Plymouth NHS Trust for their critical review of the manuscript. We are grateful for support from the Coordination for the Improvement of Higher Education Personnel (CAPES). The support and assistance offered by the Graduate Program in Epidemiology in Public Health of the National School of Public Health (ENSP) under the coordination of Leticia Oliveira Cardoso and the administrative support provided by Marcella Fagundes and Joyce Torres. Thank you all very much!

## Author Contributions

**Conceptualization:** Paulo Victor de Sousa Viana, Ana Luiza de Souza Bierrenbach, Paulo Cesar Basta.

**Data curation:** Ana Luiza de Souza Bierrenbach, Paulo Cesar Basta.

**Formal analysis:** Paulo Victor de Sousa Viana, Natalia Santana Paiva, Ana Luiza de Souza Bierrenbach, Paulo Cesar Basta.

**Methodology:** Paulo Victor de Sousa Viana, Natalia Santana Paiva, Daniel Antunes Maciel Villela, Leonardo Soares Bastos, Ana Luiza de Souza Bierrenbach, Paulo Cesar Basta.

**Supervision:** Ana Luiza de Souza Bierrenbach, Paulo Cesar Basta.

**Validation:** Daniel Antunes Maciel Villela, Leonardo Soares Bastos, Ana Luiza de Souza Bierrenbach.

**Visualization:** Ana Luiza de Souza Bierrenbach, Paulo Cesar Basta.

**Writing – original draft:** Paulo Victor de Sousa Viana, Natalia Santana Paiva, Daniel Antunes Maciel Villela, Leonardo Soares Bastos, Ana Luiza de Souza Bierrenbach, Paulo Cesar Basta.

**Writing – review & editing:** Paulo Victor de Sousa Viana, Natalia Santana Paiva, Daniel Antunes Maciel Villela, Leonardo Soares Bastos, Ana Luiza de Souza Bierrenbach, Paulo Cesar Basta.

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
