## [Decision Letter · Decision Letter 0]

23 Apr 2020

PONE-D-19-34302

Factors associated with death in patients with tuberculosis in Brazil: survival analysis with competitive risks

PLOS ONE

Dear Mr. Viana,

Thank you for submitting your manuscript to PLOS ONE. After careful consideration, we feel that it has merit but does not fully meet PLOS ONE’s publication criteria as it currently stands. Therefore, we invite you to submit a revised version of the manuscript that addresses the points raised during the review process.

We would appreciate receiving your revised manuscript by Jun 07 2020 11:59PM. To enhance the reproducibility of your results, we recommend that if applicable you deposit your laboratory protocols in protocols.io, where a protocol can be assigned its own identifier (DOI) such that it can be cited independently in the future. For instructions see: http://journals.plos.org/plosone/s/submission-guidelines#loc-laboratory-protocols

We look forward to receiving your revised manuscript.

Kind regards,

David J Horne, MD, MPH

Academic Editor

PLOS ONE

Journal Requirements:

1. In ethics statement in the manuscript and in the online submission form, please provide additional information about the patient records used in your retrospective study. Specifically, we note your statement " No informed consent was used since only the secondary notification data were

analyzed". However, it is mentioned in the text that identifying data was accessed and used; thus, please please specify whether the IRB or ethics committee waived the requirement for informed consent.

Reviewers' comments:

Reviewer's Responses to Questions

**Comments to the Author**

1. Is the manuscript technically sound, and do the data support the conclusions?

Reviewer #1: Yes

Reviewer #2: Partly

Reviewer #3: Partly

2. Has the statistical analysis been performed appropriately and rigorously? 

Reviewer #1: Yes

Reviewer #2: No

Reviewer #3: No

3. Have the authors made all data underlying the findings in their manuscript fully available?

Reviewer #1: No

Reviewer #2: Yes

Reviewer #3: No

4. Is the manuscript presented in an intelligible fashion and written in standard English?

Reviewer #1: Yes

Reviewer #2: No

Reviewer #3: Yes

5. Review Comments to the Author

Reviewer #1: This article presents an analysis of TB-related mortality in a TB-patient cohort in Brazil from 2008-2013. Reducing TB related mortality is a global goal, as TB is the greatest cause of mortality from infectious disease, so this article is of interest. The analysis links a TB database and a mortality database, which both have restrictions on public access, which is why the answer to the question above about providing underlying data with the submission was "no."

The article is generally well written, but the methods section has redundancies, is not concisely written, and is too detailed. The paper could be improved by moving some of this material to an online appendix.

I have the following additional comments that need to be addressed:

* deaths caused by TB can only be accurately assessed using autopsies. Death certificates are notorious for their errors and cannot be solely relied upon to determine cause of death. However, this analysis still has merits in the absence of autopsies. Instead of the terminology "deaths caused by TB", you should change it to be deaths probably due to TB, or something similar.

* TB diagnosed at death in patients not started on TB treatment are excluded, but this is not mentioned. This can be a substantial proportion of TB in some settings and for populations having disparities in access to care.

* For persons with HIV, the underlying cause of death on the death certificate is nearly always chosen to be HIV and not TB. This is not an error in coding, but the perception of physicians completing the forms. This is also shown in your data, as there were 209 HIV+ "deaths due to TB" and 1762 HIV+ associated TB deaths. That doesn't mean that TB did not cause their deaths (see above). How does this affect your analysis?

* Did you not have data on ART use by HIV+ during or before TB treatment? This should be a major part of the Discussion section, including mentioning of ART for HIV, and LTBI treatment among persons with HIV as a means to decrease HIV-related TB deaths, and overall TB mortality.

* in several places you write "anti-HIV serology" and in others "HIV serology". The latter is correct. And on p. 9, you have omitted the word "test" for "in progress" and "not performed".

* in the Methods, please explicitly describe the outcomes analyzed and their comparisons for each model. For typical survival analyses, the comparison is those who did not die (or a subset of those who did not die). Also include the comparison on the titles of Table 2.

* p. 11, no informed consent was "obtained" not "used"

* also p. 11 and elsewhere, interquartile range is typically abbreviated as IQR. You have it as IQ and IR in the document.

* p. 12 please present the percentages of deaths, not of cases

* Were all those who died tested for HIV? if not, how might this have biased your analysis?

* are "number of treatments" referring to separate episodes of TB disease? or of number of medications? What does this mean for deaths with no mention of TB? please describe this variable better.

* put the N at the top of Table 1, and on the columns of Table 2

* what does the category "ignored" mean for the variable "schooling"? Is this "refused to answer" or "unknown"?

* Discussion summary: please include the referent for each summary population mentioned. Also, the HR of mortality was significantly higher for all age groups beginning from 20+ compared with <20. Males were not significantly associated with TB associated deaths, so that statement is inaccurate.

* you mention "relative risks" but you analyzed hazard ratios, not relative risks.

* you discuss diabetes, but don't state that you did not find a statistically significant association with mortality. The concluding sentence of that paragraph was not shown in the study findings: "findings suggest the need for better screening of DM". What evidence do you have that DM diagnoses were missed?

* deaths early in TB treatment are common among persons with HIV. I suggest looking at the time to death, stratified by HIV, to see and present how this differs. TB prevention through LTBI treatment, and early access to HIV treatment soon after HIV infection, may be the only ways to reduce these early TB deaths.

Reviewer #2: Summary: The authors have performed a retrospective cohort study to quantify mortality among patients with TB and causes of death for patients with TB in Brazil. While this is an important study in the context of the End TB Strategy’s goal of 95% reduction in TB deaths, I think the manuscript could be significantly strengthened by addressing the major and minor comments listed below.

Major Comments

1-The abstract could be strengthened as this may be the only part of the manuscript that some will read. In the objective section, would drive home the significance with some text about TB in Brazil, End TB strategy goals, and what would be done with the results. Would include some definitions in the methods-specifically with respect to the outcome-deaths due to TB, deaths associated with TB, deaths from other causes. The conclusions are not supported by the results as active surveillance and early case finding were not assessed. Would revise the conclusions section to summarize the main findings and state what will be done with the results.

2-Would add some text about the End TB strategy in the Introduction and throughout to further highlight the importance of TB mortality studies.

3-The Methods section would benefit from some reorganization for flow. The first section could be “Study Population” in which you state the study design, setting source of study population, eligibility criteria, follow-up definition (start and stop), and ethics statement. The second section could be “Study Definitions” in which you define the TB case definition, covariates of interest, and mortality data source and mortality outcomes (death due to TB, death associated with TB, and deaths from other causes). The fourth section could describe the linkage procedures and the statistical analyses. The Strobe Checklist can be helpful with manuscript organization for observational studies (https://www.equator-network.org/wp-content/uploads/2015/10/STROBE_checklist_v4_cohort.pdf).

4-Would specifically state the TB case definition used in SINAN. Are only laboratory confirmed TB cases included or are clinical cases for which a TB treatment course is prescribed also used?

5-Why wasn’t the WHO definition of TB death considered for this study. A TB patient who dies for any reason before starting or during the course of treatment (https://apps.who.int/iris/bitstream/handle/10665/79199/9789241505345_eng.pdf;jsessionid=CE70AEF0D819CAD50145FD8211F50B55?sequence=1). This is important for comparing the results of this study with other studies.

6-In the variables of interest section it was noted that these variables have all been shown to be associated with death due to TB in previous literature. Why later was it noted that the adjusted analysis only included those with p<0.20 using the Wald test in a simple model and those with p<0.05 in a “multiple model”? Why not include all variables identified as important in the previous literature?

7-In describing Table 1 data in the text, be cautious using the term rate as these are actually frequencies and proportions. Additionally, would be careful in comparing groups as it does not appear that any statistical testing was done using the data in Table 1.

8-Figure 2 appears to be the results of the unadjusted model. It would be more illustrative to show the results of the adjusted model either in addition to or in place of the unadjusted model.

9-Was year of cohort entry considered as a variable of interest? TB screening and treatment or other important practices may have changed over the study period.

10-Were patients with documented drug resistance or patients not treated with a drug-susceptible TB regimen (HRZE) excluded? Similarly, can you somehow account adherence by looking at the time it takes the patient to complete treatment? These are both important factors when studying mortality among patients with TB.

11-The first paragraph of the Discussion jumps right to risk factors for death. The Discussion would benefit from description of the main findings of the study followed by discussion of these findings in the context of other studies conducted both within Brazil and outside Brazil.

12-The Discussion includes a paragraph about the importance of screening for DM; however, DM is not included in the adjusted model. This gets to the comment about how variables were chosen for inclusion in the multivariable model.

13-The methods section should describe how missing data were handled. Were these people excluded, was multiple imputation performed, other? What does HIV serology in progress mean?

14-The Conclusions section could be strengthened by again summarizing the main findings as well as how the results can be used to improve outcomes in Brazil (such as future studies, etc).

15-Additional limitations to consider are the presence of unmeasured confounders (smoking, chronic lung disease, etc) and generalizability of the study findings.

Minor Comments

1-Would update Reference #1 to the 2019 Global TB Report.

2-The flow chart could be made more clear by using arrows with boxes to show those excluded and those who enter due to re-classification. Would need to explicitly define the exclusions. For example, you note exclusion of non-TB cases from SINAN.

Reviewer #3: In general, this study used a national-level cohort to examine the mortality risk among people with tuberculosis. The risk factors were examined by category of deaths due to tuberculosis, associated with tuberculosis, and from other causes after the tuberculosis treatment. Several risk factors were found, with some differences by cause of death category. However, the article needs extensive editing for language and several critical points need to be clarified before considering for publication.

1. In page 5, I think it is important to provide a more complete SINAN context. For example, how does this system collect data? Who actually does the monitoring, doctor? What criteria are used to diagnoses cases? Another important thing is how do authors identify tuberculosis cases form SINAN? By using ICD code or extracting manually from medication notes? If manually, how to evaluate the reliability?

2. In page 5 and 6, the paragraph “The variable that identifies the color or race of individuals according to the categories adopted by the Brazilian Geography and Statistics Institute (IBGE)…” presents twice.

3. In page 8, what did it mean by “ii) associated TB deaths, those deaths in which there was no mention of any of the ICD-10 codes (A15-A19), referring to TB in any line of part 1 of the death certificate”? Does that mean that tuberculosis was not coded as the underlying cases (part 1), but the terminal or intermediate cause?

4. The figure 1 is confusing and cannot be matched with the “Inclusion and exclusion criteria” part totally. This part should be combined with “ Record linkage procedures and study groups definition” part and restructured.

5. In addition to reporting the mortality rate, it is more valuable to report the incidence per 100,000 patients in different follow-up time periods.

6. How did authors deal with the missing value?

7. In page 10, what is the rationale of p-value>0.20 were used to select covariates? How about adding some sensitivity analysis by adopting different cut-off p values to validate the robustness of the results?

8. The cases used in this study were patients with tuberculosis treatment. What kind of treatments did they received? The treatment information should be controlled in the model as they also ave effects on the survival time.

6. PLOS authors have the option to publish the peer review history of their article (what does this mean?). If published, this will include your full peer review and any attached files.

Reviewer #1: Yes: Suzanne Marks

Reviewer #2: No

Reviewer #3: No

---

## [Author Response · Author response to Decision Letter 0]

26 Jun 2020

Author's reply to comments:

Dear Dr. David J Horne, PLOS ONE,

First, I would like to thank you for considering our manuscript for publication in PLOS ONE. We are pleased to send the revised version of the manuscript entitled “Factors associated with death in patients with tuberculosis in Brazil: survival analysis with competitive risks” for your consideration for publication in PLOS ONE.

We also thank the reviewers for their meticulous review and appreciate the constructive criticism of our work. We received from the reviewers and hope to point out all comments appropriately. We have made a wide revision and changes to the manuscript, which are highlighted and we also addressed the comments and changes, point to point, below. It was an immense privilege to get such valuable feedback that allowed us to address several gaps in our manuscript.

In this letter, you will find our responses to each question and concern.

Kind regards, 

The authors

RESPONSES TO THE REVIEWER #1

1) This article presents an analysis of TB-related mortality in a TB-patient cohort in Brazil from 2008-2013. Reducing TB related mortality is a global goal, as TB is the greatest cause of mortality from infectious disease, so this article is of interest. The analysis links a TB database and a mortality database, which both have restrictions on public access, which is why the answer to the question above about providing underlying data with the submission was "no."

Authors comments: Thank you for your kind words; they were very well received. Your comments were invaluable in helping us improve the manuscript. In Brazil, there is a Law, known as the Access to Information Law, which protects the confidentiality of patients' information reported in their information systems. Thus, for the linkage of the databases, we researchers signed a document making us responsible for the integrity of the information made available. The individual data sets of tuberculosis cases and deaths in Brazil analyzed during the present study are available upon request in the repository “Electronic System of the Citizen Information Service”, [http://esic.cgu.gov.br/sistema/ site / index.html? ReturnUrl =% 2fsystem].

2) The article is generally well written, but the methods section has redundancies, is not concisely written, and is too detailed. The paper could be improved by moving some of this material to an online appendix.

Authors comments: We are grateful for your comments about the importance of the methods of study. We clarified the linkage process section in the supplementary file.

3) Deaths caused by TB can only be accurately assessed using autopsies. Death certificates are notorious for their errors and cannot be solely relied upon to determine cause of death. However, this analysis still has merits in the absence of autopsies. Instead of the terminology "deaths caused by TB", you should change it to be deaths probably due to TB, or something similar.

Authors comments: We agree with the reviewer that the gold standard for death certifications is autopsies. Throughout the manuscript, we replaced the term "death due TB" with " probable TB deaths ".

4) TB diagnosed at death in patients not started on TB treatment are excluded, but this is not mentioned. This can be a substantial proportion of TB in some settings and for populations having disparities in access to care.

Authors comments: We agree with the reviewer that we were unable to capture the underreporting of TB cases from death certificates (SIM). This limitation was included in the discussion section, such weakness. Due to some restrictions that we had faced to the national death database in Brazil, the Brazilian government only provided us with the deaths that have already linked with the notification cases database, i.e. which were under treatment (Sinan).

5) For persons with HIV, the underlying cause of death on the death certificate is nearly always chosen to be HIV and not TB. This is not an error in coding, but the perception of physicians completing the forms. This is also shown in your data, as there were 209 HIV+ "deaths due to TB" and 1762 HIV+ associated TB deaths. That doesn't mean that TB did not cause their deaths (see above). How does this affect your analysis?

Authors comments: We appreciate your comments on the relevance of this placement. Such concern was discussed as limitations in the discussion section of the article, regarding the aspect of coding underlying causes in HIV positive patients.

6) Did you not have data on ART use by HIV+ during or before TB treatment? This should be a major part of the Discussion section, including mentioning of ART for HIV, and LTBI treatment among persons with HIV as a means to decrease HIV-related TB deaths, and overall TB mortality.

Authors comments: Unfortunately, in the Brazilian compulsory notification system (Sinan), we do not have information on ART. We discussed the limitations of this information among HIV-positive patients in our manuscript. In addition, we have included a discussion on the identification and treatment of latent TB infection in people living with HIV.

7) In several places you write "anti-HIV serology" and in others "HIV serology". The latter is correct. And on p. 9, you have omitted the word "test" for "in progress" and "not performed".

Authors comments: We appreciate your important observation in terminology. We conducted a comprehensive review of the terms and all have been replaced for “HIV serology”.

8) In the Methods, please explicitly describe the outcomes analyzed and their comparisons for each model. For typical survival analyses, the comparison is those who did not die (or a subset of those who did not die). Also, include the comparison on the titles of Table 2.

Authors comments: We appreciate your observation. In the methods section, we inserted the comparison categories for each outcome compared to the censoring, as well as introducing this information in table 2.

9) p. 11, no informed consent was "obtained" not "used"

Author comments: Thanks for your observation, we changed the word to obtained

10) Also p. 11 and elsewhere, interquartile range is typically abbreviated as IQR. You have it as IQ and IR in the document.

Author comments: Thanks for your observation, we changed the abbreviation to IQR

11) p. 12 please present the percentages of deaths, not of cases

Author comments: We corrected and presented the percentages for deaths in the results section

12) Were all those who died tested for HIV? if not, how might this have biased your analysis?

Author comments: Although the main international recommendations are to test all cases for HIV serology, unfortunately in Brazil due to their continental geographical characteristics, many regions are unable to test their patients satisfactorily for various reasons, that include the poor organization of the health care network, the lack of HIV tests and the not awareness of the health workers. This point was also addressed in the discussion of the study.

13) Are "number of treatments" referring to separate episodes of TB disease? or of number of medications? What does this mean for deaths with no mention of TB? please describe this variable better.

Author comments: This variable reports the number of times that a patient started a new treatment for tuberculosis during the follow-up, for example, the same patient who was a new case in 2008, re-entered in the system at other times to start a new treatment in later years, whether due to recurrence or by re-entry after abandoning treatment. As for deaths with no mention of tuberculosis, this points to an important weakness of health services that are unaware that the patient was undergoing treatment for tuberculosis and end up not reporting in the death certificate.

14) Put the N at the top of Table 1, and on the columns of Table 2

Author comments: Thanks for the comment, information placed in the tables

15) What does the category "ignored" mean for the variable "schooling"? Is this "refused to answer" or "unknown"?

Author comments: They were included in the category ignored cases with absence of information and children under 5 years

16) Discussion summary: please include the referent for each summary population mentioned. Also, the HR of mortality was significantly higher for all age groups beginning from 20+ compared with <20. Males were not significantly associated with TB associated deaths, so that statement is inaccurate

Authors comments: We appreciate the concern in structuring the summary of the discussion and restructured the first paragraph better.

17) * you mention "relative risks" but you analyzed hazard ratios, not relative risks.

Authors comments: Thank you very much for the observation we corrected the mistakes and replaced the term “relative risk” for “hazard ratio”.

18) You discuss diabetes, but don't state that you did not find a statistically significant association with mortality. The concluding sentence of that paragraph was not shown in the study findings: "findings suggest the need for better screening of DM". What evidence do you have that DM diagnoses were missed?

Authors comments: In our study, we propose the hypothesis that diabetes could contribute to the mortality in patients undergoing treatment for tuberculosis. But, in fact, this variable did not show statistical significance in our analyzes. Therefore, we are not be able to evaluate if there was underdiagnoses of diabetes in the TB patients, in Brazil, and due to this, following your recommendations we removed that sentence from the discussion.

19) Deaths early in TB treatment are common among persons with HIV. I suggest looking at the time to death, stratified by HIV, to see and present how this differs. 

Authors comments: We included a figure in the Supporting Information showing the survival curves stratified by HIV serology.

20) TB prevention through LTBI treatment, and early access to HIV treatment soon after HIV infection, may be the only ways to reduce these early TB deaths.

Authors comments: We thank the reviewer for this important statement. In fact, the early initiation of antiretroviral therapy and the early identification of latent TB infection is an important way to reduce mortality in this group. We’ve included an excerpt addressing this in the discussion section.

RESPONSES TO THE REVIEWER #2

Major Comments

1) The abstract could be strengthened as this may be the only part of the manuscript that some will read. In the objective section, would drive home the significance with some text about TB in Brazil, End TB strategy goals, and what would be done with the results. Would include some definitions in the methods-specifically with respect to the outcome-deaths due to TB, deaths associated with TB, deaths from other causes. The conclusions are not supported by the results as active surveillance and early case finding were not assessed. Would revise the conclusions section to summarize the main findings and state what will be done with the results.

Authors comments: We appreciate your important contributions in the structure of the abstract of the manuscript. We improved the summary according to your comments.

2) Would add some text about the End TB strategy in the Introduction and throughout to further highlight the importance of TB mortality studies.

Authors comments: We included a sentence in the introduction to the manuscript describing the End TB strategy

3)The Methods section would benefit from some reorganization for flow. 

The first section could be “Study Population” in which you state the study design, setting source of study population, eligibility criteria, follow-up definition (start and stop), and ethics statement. 

The second section could be “Study Definitions” in which you define the TB case definition, covariates of interest, and mortality data source and mortality outcomes (death due to TB, death associated with TB, and deaths from other causes). 

The fourth section could describe the linkage procedures and the statistical analyses. The Strobe Checklist can be helpful with manuscript organization for observational studies (https://www.equator-network.org/wp-content/uploads/2015/10/STROBE_checklist_v4_cohort.pdf

Authors comments: We agree with your valuable comments and we replaced the structure of the methods section, according your suggestions.

4) Would specifically state the TB case definition used in SINAN. Are only laboratory confirmed TB cases included or are clinical cases for which a TB treatment course is prescribed also used?

Authors comments: We have included in the methods section a brief description of the case definition of tuberculosis, used in SINAN database in Brazil. By this criterion, we use not only bacteriologically confirmed TB case and clinically diagnosed TB case, as proposed by WHO definition, but also cases with epidemiological link to other confirmed TB cases. 

5) Why wasn’t the WHO definition of TB death considered for this study. A TB patient who dies for any reason before starting or during the course of treatment (https://apps.who.int/iris/bitstream/handle/10665/79199/9789241505345_eng.pdf;jsessionid=CE70AEF0D819CAD50145FD8211F50B55?sequence=1). This is important for comparing the results of this study with other studies.

Authors comments: According to cited document by the reviewer, for country-specific purposes, deaths due to TB and deaths due to other causes could be separated in the treatment outcomes section. However, the two need to be added together for treatment outcome monitoring.

In spite of we have studied different kinds of mortality among patients under TB treatment, for the SINAN database all case have had death as an outcome of treatment were counted for treatment outcome monitoring.

6) In the variables of interest section it was noted that these variables have all been shown to be associated with death due to TB in previous literature. Why later was it noted that the adjusted analysis only included those with p<0.20 using the Wald test in a simple model and those with p<0.05 in a “multiple model”? Why not include all variables identified as important in the previous literature?

Authors comments: We appreciate the concern of the reviewer, in our study, although we selected an already known group of variables associated with death, this set of variables was still relatively large. Therefore, with the concern of being as flexible as possible and based on other studies in the literature, we also decided to adopt a cut-off point of p-value of 0.20 in order to seek a more parsimonious model as possible to describe the data, which also results better numerical stability and generalization of results. Thus, to better clarify the process of selecting variables for the Fine & Gray regression model, we rewrite the following sentence in the methods section.

“In our study, based on a group of variables associated with TB death in the literature, we based their choice individually, verifying the significance of each variable and the elimination of non-significant variables. Any variable with a p-value of 0.20 in the univariate test was selected as a candidate for the Fine & Gray multiple model, and the level of significance for choosing the final model was 5%”.

7) In describing Table 1 data in the text, be cautious using the term rate as these are actually frequencies and proportions. Additionally, would be careful in comparing groups as it does not appear that any statistical testing was done using the data in Table 1.

Authors comments: We are grateful for the reviewer's comment, in fact analyzes of proportions were performed in table 1 and not rates. We corrected this mistake throughout the results section.

8) Figure 2 appears to be the results of the unadjusted model. It would be more illustrative to show the results of the adjusted model either in addition to or in place of the unadjusted model.

Authors comments: In fact, in figure 2, the survival curves for the unadjusted data are presented in an exploratory way to better guide our analyzes in the Fine & Gray regression model.

9) Was year of cohort entry considered as a variable of interest? TB screening and treatment or other important practices may have changed over the study period?

Authors comments: In our study, the year of entry into the cohort was not considered an important variable for analysis. In Brazil, during our analysis period, there was no change in the type of treatment of tuberculosis cases.

10) Were patients with documented drug resistance or patients not treated with a drug-susceptible TB regimen (HRZE) excluded? Similarly, can you somehow account adherence by looking at the time it takes the patient to complete treatment? These are both important factors when studying mortality among patients with TB.

Authors comments: In our study, we excluded resistant cases from the study, as they are followed up in another information system focused exclusively on drug-resistant tuberculosis cases.

11) The first paragraph of the Discussion jumps right to risk factors for death. The Discussion would benefit from description of the main findings of the study followed by discussion of these findings in the context of other studies conducted both within Brazil and outside Brazil.

Authors comments: We thank you for your important contribution. We inform that we included a brief description with the main results of the study, in the first paragraph of the discussion.

12) The Discussion includes a paragraph about the importance of screening for DM; however, DM is not included in the adjusted model. This gets to the comment about how variables were chosen for inclusion in the multivariable model.

Authors comments: In our study, we propose the hypothesis that diabetes could contribute to the mortality in patients undergoing treatment for tuberculosis. But, in fact, this variable did not show statistical significance in our analyzes. Therefore, we are not be able to evaluate if there was underdiagnoses of diabetes in the TB patients, in Brazil, and due to this, following your recommendations we removed that sentence from the discussion.

13) The methods section should describe how missing data were handled. Were these people excluded, was multiple imputation performed, other? What does HIV serology in progress mean?

Authors comments: In our study, the missing data in the categories of variables were not excluded, but classified as ignored to avoid loss of information in the study. Thus, we do not adopt multiple imputation. When a specific exam is not collected, health professionals tend not to fill in the variable, leaving it with missing data. That is why, we believe that the missing data are completely random, the incompleteness is not related to exposure, covariates or the outcome.

In Brazil, serology in progress means that the patient has been tested but is still waiting for a positive or negative result. Unfortunately, due to logistic or operational issues at the local health units, often some patients have never get access these results.

14) The Conclusions section could be strengthened by again summarizing the main findings as well as how the results can be used to improve outcomes in Brazil (such as future studies, etc).

Authors comments: We appreciate your important contribution and adjust the conclusion according to your suggestions.

It is the first time that databases of TB cases and of mortality, from a nationwide cohort, in Brazil, were used to investigate associated factors to death in patients under treatment for TB, considering the presence of competitive events in a survival analysis. Usually, authors performing survival analysis using only two categories, deaths due to TB and deaths for other causes. In this study, we consider other deaths associated with TB and its associated factors as a third category of analysis.

Our findings indicate that the main factors associated with deaths, regardless of the cause, include age group, schooling, mixed clinical form, HIV serology and alcoholism. 

Considering age-groups, there was a dose-response effect in the risk of mortality from 20 years old, reaching the remarkable risk in patients over than 60, especially in the deaths for other causes and probable TB deaths, respectively. Taking into account the schooling, patients that were illiterate presented the highest risk of mortality, being the effect more evident to probable TB deaths. The association of mixed clinical form with all kind of deaths, likely, is result of the clinical severity of the cases, especially among those where the beginning of the treatment and the probable TB deaths occurred in short time lapse. When we focus on HIV serology, the data are contradictory. While, HIV serology positive was strongly related to TB-related deaths and to other causes of deaths, for probable TB deaths the HIV serology in progress and not performed have presented strong association. Perhaps, these inconclusive HIV serology tests acted as an indicator of the bad performance of the health services. Finally, alcoholism was associated with all forms of mortality, being stronger among probable TB deaths.

Considering the factors associated with probable TB deaths, in addition to the factors above mentioned, the following stand out: male gender, "black" and "brown" color or race, and the situation in the South region of the country. Among the TB-related deaths, in addition to the factors above mentioned, we highlighted color or race "black" and "brown" and the situation in the South and Central-West regions of the country. In turn, among the deaths from other causes, we underlined male sex, the situation in the South and Central-West regions, as well extrapulmonary clinical forms.

15) Additional limitations to consider are the presence of unmeasured confounders (smoking, chronic lung disease, etc) and generalizability of the study findings.

Authors comments: We appreciate your observation; we have included these questions in the discussion section of the manuscript.

Minor Comments

1) Would update Reference #1 to the 2019 Global TB Report.

Authors comments: We updated this bibliographic reference of the Global Report

2) The flow chart could be made clearer by using arrows with boxes to show those excluded and those who enter due to re-classification. Would need to explicitly define the exclusions. For example, you note exclusion of non-TB cases from SINAN.

Authors comments: We improved figure 1 flowchart as suggested.

RESPONSES TO THE REVIEWER #3

1) In page 5, I think it is important to provide a more complete SINAN context. For example, how does this system collect data? Who actually does the monitoring, doctor? What criteria are used to diagnoses cases? Another important thing is how do authors identify tuberculosis cases form SINAN? By using ICD code or extracting manually from medication notes? If manually, how to evaluate the reliability?

Authors comments: In the methods section of the manuscript we include a description of how Sinan information is operated, as well as its filling and analysis routine.

2) In page 5 and 6, the paragraph “The variable that identifies the color or race of individuals according to the categories adopted by the Brazilian Geography and Statistics Institute (IBGE)…” presents twice.

Authors comments: Thank you for your observation and perform the correction in the manuscript.

3) In page 8, what did it mean by “ii) associated TB deaths, those deaths in which there was no mention of any of the ICD-10 codes (A15-A19), referring to TB in any line of part 1 of the death certificate”? Does that mean that tuberculosis was not coded as the underlying cases (part 1), but the terminal or intermediate cause?

Authors comments: Yes, tuberculosis was not a cause of death, but an intermediate or even terminal cause

4) The figure 1 is confusing and cannot be matched with the “Inclusion and exclusion criteria” part totally. This part should be combined with “Record linkage procedures and study groups definition” part and restructured.

Authors comments: We appreciate your important placement. This information was included in the supplementary section of the manuscript describing the database linkage process.

5) How did authors deal with the missing value?

Authors comments: In our study, the missing data in the categories of variables were not excluded, but classified as ignored to avoid loss of information in the study. Thus, we do not adopt multiple imputation. When a specific exam is not collected, health professionals tend not to fill in the variable, leaving it with missing data. That is why, we believe that the missing data are completely random, the incompleteness is not related to exposure, covariates or the outcome.

6) In page 10, what is the rationale of p-value>0.20 were used to select covariates? How about adding some sensitivity analysis by adopting different cut-off p values to validate the robustness of the results?

Authors comments: We appreciate the concern of the reviewer, in our study, although we selected an already known group of variables associated with death, this set of variables was still relatively large. Therefore, with the concern of being as flexible as possible and based on other studies in the literature, we also decided to adopt a cut-off point of p-value of 0.20 in order to seek a more parsimonious model as possible to describe the data, which also results better numerical stability and generalization of results. Thus, to better clarify the process of selecting variables for the Fine & Gray regression model, we rewrite the following sentence in the methods section.

“In our study, based on a group of variables associated with TB death in the literature, we based their choice individually, verifying the significance of each variable and the elimination of non-significant variables. Any variable with a p-value of 0.20 in the univariate test was selected as a candidate for the Fine & Gray multiple model, and the level of significance for choosing the final model was 5%”.

7) The cases used in this study were patients with tuberculosis treatment. What kind of treatments did they received? The treatment information should be controlled in the model as they also have effects on the survival time.

Authors comments: Unfortunately, in our database of Sinan there was no information on the type of drugs used in the treatment for tuberculosis, this can be explained that TB cases in Brazil receive a single standardized treatment.

---

## [Decision Letter · Decision Letter 1]

4 Aug 2020

PONE-D-19-34302R1

Factors associated with death in patients with tuberculosis in Brazil: survival analysis with competitive risks

PLOS ONE

Dear Dr. Viana,

Thank you for submitting your manuscript to PLOS ONE. After careful consideration, we feel that it has merit but does not fully meet PLOS ONE’s publication criteria as it currently stands. Therefore, we invite you to submit a revised version of the manuscript that addresses the points raised during the review process.

We look forward to receiving your revised manuscript.

Kind regards,

David J Horne, MD, MPH

Academic Editor

PLOS ONE

Additional Editor Comments (if provided):

In addition to the comments below, Reviewer 2 had previously requested that reference 1 be updated to the 2019 WHO Global Tuberculosis Report. Although the citation was updated, the language in the introduction was not. You still refer to 2017 deaths from TB -- the 2019 report is on deaths in 2018. Please update the year and the accurate number of deaths for 2018.

Reviewers' comments:

Reviewer's Responses to Questions

**Comments to the Author**

1. If the authors have adequately addressed your comments raised in a previous round of review and you feel that this manuscript is now acceptable for publication, you may indicate that here to bypass the “Comments to the Author” section, enter your conflict of interest statement in the “Confidential to Editor” section, and submit your "Accept" recommendation.

Reviewer #3: (No Response)

2. Is the manuscript technically sound, and do the data support the conclusions?

Reviewer #3: Yes

3. Has the statistical analysis been performed appropriately and rigorously? 

Reviewer #3: Yes

4. Have the authors made all data underlying the findings in their manuscript fully available?

Reviewer #3: No

5. Is the manuscript presented in an intelligible fashion and written in standard English?

Reviewer #3: No

6. Review Comments to the Author

Reviewer #3: Although the authors have addressed the majority of my previous comments, some issues remain.

Some terms adopted for presenting the statistical methods need to be revised.

I think “survival analysis with competitive risk” is a rarely used term. “survival regression model” is also strange. It should be Cox regression/Cox proportional hazard model, or in your context, Fine and Gray competing risk regression model. Please go through all statistical terms again and make sure that they are standard terms commonly used in literature.

Besides, expressions such as “we believe”, “databases are fed continuously” should not be in academic writing.

Regarding my previous comments about a more detailed description of the data source, the sample representativeness is still not entirely clear to me. I don’t think “good coverage throughout the country” is precise enough. As this is critical for evaluating the external validity of the results, please try to add more accurate information.

Regarding the response about how missing values were handled. I think using complete cases is not a big issue. The important thing is to report the procedure - how the final sample size was arrived at (see the STROBE checklist). Even if you did not impute the data (which sometimes, indeed, may create another set of problems), the number of participants involved in each exclusion step needs to be reported (a flow chart is helpful, particularly for cohort study using electronic health records).

7. PLOS authors have the option to publish the peer review history of their article (what does this mean?). If published, this will include your full peer review and any attached files.

Reviewer #3: No

---

## [Author Response · Author response to Decision Letter 1]

17 Sep 2020

Author's reply to comments:

Dear Dr. David J Horne, PLOS ONE,

First, I would like to thank you for considering our manuscript for publication in PLOS ONE. We are pleased to send the revised version of the manuscript entitled “Factors associated with death in patients with tuberculosis in Brazil: survival analysis with competitive risks” for your consideration for publication in PLOS ONE.

We also thank the reviewers for their meticulous review and appreciate the constructive criticism of our work. We received from the reviewers and hope to point out all comments appropriately. We have made a wide revision and changes to the manuscript, which are highlighted and we also addressed the comments and changes, point to point, below. It was an immense privilege to get such valuable feedback that allowed us to address several gaps in our manuscript.

In this letter, you will find our responses to each question and concern.

Kind regards, 

The authors

Additional Editor Comments (if provided):

In addition to the comments below, Reviewer 2 had previously requested that reference 1 be updated to the 2019 WHO Global Tuberculosis Report. Although the citation was updated, the language in the introduction was not. You still refer to 2017 deaths from TB -- the 2019 report is on deaths in 2018. Please update the year and the accurate number of deaths for 2018.

Authors comments: We appreciate the observation, we have corrected the epidemiological data of the global tuberculosis report for the year 2018.

RESPONSES TO THE REVIEWER #3

1) Some terms adopted for presenting the statistical methods need to be revised.

I think “survival analysis with competitive risk” is a rarely used term. “survival regression model” is also strange. It should be Cox regression/Cox proportional hazard model, or in your context, Fine and Gray competing risk regression model. Please go through all statistical terms again and make sure that they are standard terms commonly used in literature.

Authors comments: We appreciate your important placement. We carried out an extensive review and correction using the same nomenclatures used by one of the creators of the method (the researcher JP Fine), in a recent manuscript "Practical recommendations for reporting Fine-Gray model analyzes for competing risk data", for more details: https: / /pubmed.ncbi.nlm.nih.gov/28913837/

2) Besides, expressions such as “we believe”, “databases are fed continuously” should not be in academic writing.

Authors comments: Thank you for your observation and we have corrected these expressions in the manuscript.

3) Regarding my previous comments about a more detailed description of the data source, the sample representativeness is still not entirely clear to me. I don’t think “good coverage throughout the country” is precise enough. As this is critical for evaluating the external validity of the results, please try to add more accurate information.

Authors comments: We appreciate your important placement. More information was included in the aforementioned manuscript that describes the data source used to link the records, reporting their potential and also limitations of use in Brazil.

4) Regarding the response about how missing values were handled. I think using complete cases is not a big issue. The important thing is to report the procedure - how the final sample size was arrived at (see the STROBE checklist). Even if you did not impute the data (which sometimes, indeed, may create another set of problems), the number of participants involved in each exclusion step needs to be reported (a flow chart is helpful, particularly for cohort study using electronic health records).

Authors comments: We are grateful for the reviewer's concern and when re-evaluating the figure of the linkage process presented, it was confusing, as we mixed reclassification information of "treatment status" and exclusion criteria for observations (duplicate observations, inconsistencies in dates, among others). We redid the flowchart only with the information on the exclusions of the observations according to the exclusion criteria of the study cohort.

---

## [Editor Report · Decision Letter 2]

21 Sep 2020

Factors associated with death in patients with tuberculosis in Brazil: competing risks analysis

PONE-D-19-34302R2

Dear Dr. Viana,

We’re pleased to inform you that your manuscript has been judged scientifically suitable for publication and will be formally accepted for publication once it meets all outstanding technical requirements.

Kind regards,

David J Horne, MD, MPH

Academic Editor

PLOS ONE
---

## [Editor Report · Acceptance letter]

23 Sep 2020

PONE-D-19-34302R2 

Factors associated with death in patients with tuberculosis in Brazil: competing risks analysis 

Dear Dr. Viana:

I'm pleased to inform you that your manuscript has been deemed suitable for publication in PLOS ONE. Congratulations! Your manuscript is now with our production department. 

Kind regards, 

on behalf of

Dr. David J Horne 

Academic Editor

PLOS ONE